# Effect of Nasogastric Tube Placement, Manipulation, and Fluid Administration on Transcutaneous Ultrasound Visualization and Assessment of Stomach Position in Healthy Unfed and Fed Horses

**DOI:** 10.3390/ani12233433

**Published:** 2022-12-06

**Authors:** Kira Lyn Epstein, Mark David Hall

**Affiliations:** 1Department of Large Animal Medicine, College of Veterinary Medicine, University of Georgia, Athens, GA 30606, USA; 2Bain and Company, London WC2N 5RW, UK

**Keywords:** gastric, stomach, ultrasound, horse, nasogastric tube

## Abstract

**Simple Summary:**

In horses with colic (abdominal pain), it is vital to be able to identify and remove fluid building up in their stomachs. Ultrasound is a non-invasive diagnostic that is often used to estimate the size of the stomach in horses with colic, but our knowledge of ultrasound of the stomach in horses with colic is not complete. Because horses with colic may or may not be eating and a veterinarian often passes a stomach tube to help remove or give fluid, we need to know how feeding and the use of a stomach tube affects ultrasound of the stomach to help veterinarians interpret ultrasound of the stomach in horses with colic appropriately. In a group of healthy horses, ultrasonography revealed that feeding, stomach tube placement, and giving fluid, increased the size of the stomach. After fluid was given, it could be consistently identified within the stomach with ultrasound. Based on our findings, if a horse has been eating recently or had a stomach tube passed, the size of the stomach on ultrasound may be increased, unrelated to fluid buildup, and looking for fluid in the stomach may be a better method to diagnose fluid building up.

**Abstract:**

Knowledge of the effects of feeding and nasogastric tube placement and manipulation on gastric ultrasound is limited. Given the variability in duration since feeding and the ubiquitous use of nasogastric tubes in horses with colic, the interpretation of gastric ultrasound in horses with colic requires an understanding of these effects. Cranial to caudal and dorsal to ventral ultrasonographic dimensions of the stomach were obtained in 10 unfed horses and five fed horses, before and after nasogastric tube placement, after checking for reflux and after administration of 6 L of water in unfed horses. Fed horses’ stomachs were larger in both cranial to caudal and dorsal to ventral dimension than unfed horses. Nasogastric intubation and the administration of water increased ultrasonographic gastric dimensions in fed and unfed horses. Checking for reflux did not consistently decrease ultrasonographic gastric dimension in fed or unfed horses. Fluid was consistently identified in the stomach with ultrasound after 6 L of water. Increases in gastric ultrasound dimensions found in horses that have been recently fed and/or had a nasogastric tube placed can occur without pathologic gastric distension related to colic and should be interpreted in this context. In contrast, the identification of fluid in the stomach on ultrasound occurs consistently with fluid administration and may be more useful than standard ultrasound parameters of gastric dimensions to identify horses with colic likely to have significant gastric reflux.

## 1. Introduction

Transabdominal ultrasound has become an important diagnostic tool in horses with acute abdominal pain, since being introduced over 30 years ago [1,2]. During colic workup, clinicians may elect to perform a systematic ultrasound examination of the entire abdomen [3,4] or chose to perform a more rapid, targeted examination for horses with colic, similar to the fast localized abdominal sonography of horses (FLASH) protocol described by Busoni et al. [5]. Additionally, there appears to be a role for ultrasonographic monitoring during the treatment of horses with colic [6,7,8,9]. An evaluation of the stomach size is a common exam component of both complete and targeted diagnostic transabdominal ultrasound examinations, and may have a role in monitoring for gastric reflux accumulation during treatment [3,4,5,6,10]. Although primary gastric disease is uncommon in horses, secondary distension of the stomach is an important diagnostic clue that can be painful and life-threatening. Because fluid accumulation in the stomach most frequently occurs secondary to functional (ileus and inflammatory (anterior enteritis) diseases) or mechanical (non-strangulating or strangulating) obstructions of the small intestine, the presence of increased fluid in the stomach is highly suggestive of a disease within the small intestine [11]. 

The identification and removal of the accumulating fluid is vital to prevent fatal gastric rupture, as well as decrease pain at initial presentation and during treatment, particularly in the case of anterior enteritis and post-operative reflux and/or ileus. Nasogastric intubation is the standard method for the diagnosis and treatment of the accumulation of gastric fluid. However, nasogastric intubation does not always result in complete evacuation of the stomach and there are reports of gastric rupture with a nasogastric tube in place [12]. Additionally, the use of either long-term indwelling or the repeated passage of a nasogastric tube can be uncomfortable, require repeated sedation, and/or have the potential to cause complications such as decreased gastric emptying [13] or pharyngeal damage [14]. Ideally, ultrasound could be used to determine if gastric contents are accumulating, requiring nasogastric intubation, and if gastric decompression via nasogastric tube is complete. Experimental studies in healthy horses and ponies [15,16], and a single clinical study of horses with gastric reflux [6], suggest that gastric distension and fluid accumulation may be able to be detected and monitored using ultrasound.

Like any diagnostic tool, the effective use of ultrasound requires a clear definition of normal, variations of normal, and expected changes with common interventions and pathologies. Despite the frequency of inclusion of ultrasound evaluation of the stomach in horses with colic, not all of this information is well known. The anatomic location, size, and gas within the contents of the stomach prevent complete imaging of the stomach. As a result, options for evaluating the size of the stomach using ultrasound are based on inference from the dimensions (cranial to caudal and dorsal to ventral) of the portion of the stomach that can be seen on the left side. In the cranial to caudal dimension, the stomach size is referenced to the intercostal spaces. It should be noted that in studies and descriptions of normal horses, there is inconsistency in which specific intercostal spaces and the number of intercostal spaces that the stomach can be imaged [10,17,18]. Additionally, differences between foals [19], ponies [20], and adult horses have been identified. Dorsal to ventral location and measurements of the stomach have not been consistently reported in descriptive studies of normal foals, adult horses, or ponies. In healthy horses and ponies, feed [21], air [15], and fluid [16], have been shown to affect the cranial to caudal and dorsal to ventral location and the measurement of the stomach with ultrasound. Sedation with xylazine did not appear to affect gastric ultrasound [21]. However, morphine administration resulted in being able to image the stomach more caudally [22].

If gastric ultrasound is to be used for diagnosis and monitoring in horses with colic, the effect of having a nasogastric tube in place and using it to check for reflux is key to interpretation given the overlap in the indications for their use. The authors’ experience suggests that placement and/or maintenance of a nasogastric tube in horses can affect gastric ultrasound. This is supported by a study where nasogastric tubes were left in place for 30–80 min in unfed horses and the stomach moved from a ventral location to a more “normal” location, dorsal to the costochondral junctions [21]. This study was not focused on gastric ultrasound and provided limited details and no statistical comparisons. Additionally, the nasogastric tube was not manipulated, no fluid was administered, and all horses were unfed because the authors felt that nasogastric intubation in fed horses was unlikely to be clinically relevant. While often horses have been off feed when they have colic, this is not always the case, particularly when evaluated early on in the course of the disease.

This study aims to add to the understanding of gastric ultrasound findings in horses with colic. Specifically, this study builds upon previous studies by combining several interventions that have been previously evaluated in isolation and adds manipulation of the nasogastric tube. Under each condition similar parameters will be evaluated to allow qualitative comparisons between interventions, as well as statistical comparisons of quantitative measurements and qualitative assessments between interventions. The objectives of this study are to describe and compare the effects of nasogastric intubation and checking for gastric reflux with a nasogastric tube in fed and unfed horses, and describe the effects of nasogastric administration of fluid in unfed healthy horses on gastric ultrasound. We hypothesized that feeding, nasogastric intubation, checking for nasogastric reflux, and the administration of fluid via the nasogastric tube would result in increases in the cranial to caudal and dorsal to ventral ultrasound dimensions of the stomach.

## 2. Materials and Methods

### 2.1. Animals

The study was performed using 10 University owned horses (8 geldings and 2 mares; 4 thoroughbreds, 5 quarter horses, and 1 warmblood; age 9–23 years; weight 420–545 kg). The horses were clinically healthy based on physical examination. They had no known history of gastrointestinal disease and no signs of gastrointestinal problems were detected on physical examination.

### 2.2. Study Design

Horses were fed a diet of ad libitum grass hay supplemented with approximately 1 pound of pelleted compete feed twice daily. All horses were allowed free access to water throughout the study. In the first trial (unfed), horses were not fed for 8–12 h overnight prior to the study. Ultrasounds were performed prior to feeding the following morning. Some, but not all horses were muzzled to prevent food or shavings consumption overnight. Ultrasound examination of the stomach was performed at 4 timepoints: prior to nasogastric tube (NGT) placement for baseline information (pre-NGT), within 5 min of NGT placement (post-NGT), within 5 min of checking for reflux with the NGT (post-reflux), and within 5 min of administration of 6 L of water NGT (post-water).

NGT placement and manipulation was facilitated by administration of 0.28–0.69 mg/kg xylazine intravenously as needed. NGT placement was performed by an experienced veterinarian (one of the investigators) through either the left or right nasal passage. The NGT was passed until it could not be advanced further easily so that the tube was well seated within the stomach. Checking for reflux was performed by priming the tube with 2 L of water to establish a siphon followed by manipulating the tube while applying suction with a dose syringe to simulate the practice used clinically in horses with colic. If reflux was obtained the quantity was recorded. Horses were administered a total of 6 L of water inclusive of the water used to check for reflux.

In the second trial (fed), 5 of the horses (4 geldings and 1 mare; 1 thoroughbred, 3 quarter horses, and 1 warmblood; age 15–23 years; weight 490–545 kg) were fed normally throughout the study. The amount of feed consumed and if horses were eating at the time the trial started were not recorded. It was common for horses to be eating at the time they were examined just prior to sedation for NGT placement. There were at least 24 h between the first and second trial. Ultrasound examinations and procedures for the first trial were repeated with the exception of the administration of 6 L of water to avoid any risk of overdistension of the stomach in fed horses.

### 2.3. Ultrasound Examinations

Ultrasound examinations were performed with a portable ultrasound machine with a 2–5 MHz curved array probe (Sonosite Micromaxx; FUJIFILM Sonosite, Inc; Bothell, WA, USA) at a depth of 20 cm. A systematic ultrasound exam of the stomach was performed using isopropyl alcohol to soak the hair to facilitate adequate contact of the transducer. Identification of the stomach was based on visualization of a characteristic semicircular gastrointestinal structure adjacent to the spleen and splenic vein [23]. Each ultrasound was attended by two veterinarians experienced with abdominal ultrasound in horses (both investigators).

The exam was started in the left 12th intercostal space (ICS). The 12th intercostal space was identified by counting cranially from the last ICS (17th) and marked with a piece of tape at the level of the point of the elbow (tuber olecrani). Within the ICS, the abdomen was imaged from the ventral border of the lung field where the diaphragm could be visualized to the most ventral aspect of the rib space. ICS cranial to the 12th were scanned to identify the cranial extent of the stomach as far cranial as the 4th ICS. The most cranial ICS where the stomach could be visualized was recorded (most cranial ICS). Then, ICS caudal to the 12th were scanned to identify the caudal extent of the stomach as far caudal as the 17th ICS. The most caudal ICS where the stomach could be visualized was recorded (most caudal ICS). The cranial to caudal length of the stomach was determined by counting the number of ICS where the stomach could be visualized from the most cranial to the most caudal including these spaces (number of ICS spanned). If the stomach was identified at the 12th ICS, the distance from the tape (level of the point of the elbow) to the most dorsal location the stomach could be identified was measured in centimeters (height of stomach at ICS 12). In the first trial, after administration of 6 L of water the most cranial and most caudal ICS where fluid (hypoechoic contents) could be identified in the stomach were recorded.

### 2.4. Data Collected

Ultrasound parameters for the stomach that were collected from each ultrasound examination included if the stomach could be identified on ultrasound, the cranial to caudal dimension of the stomach defined by the most cranial ICS, most caudal ICS, and number of ICS (inclusive) where the stomach could be identified with ultrasound, and the dorsal to ventral dimension of the stomach defined by the height of stomach at ICS 12. The change in the height of the stomach at ICS from previous timepoint(s) was calculated and recorded. For ultrasound examinations performed after administration of 6 L of water via NGT, the most cranial ICS, most caudal ICS, and ICS spanned where fluid within the stomach could be identified was also recorded.

Qualitative comparisons between timepoints (pre-NGT, post-NGT, post-reflux, and post-water) were recorded. The stomach was classified as being imaged more, unchanged, or less cranial or caudal and in more, unchanged, or less ICS than in previous timepoint(s). Further, the stomach was classified as being or not being (binary) more cranial, more caudal, or in more ICS that in previous timepoint(s). The height of the stomach at ICS 12 was classified as increased, no change, or decreased compared to previous timepoint(s). For all qualitative comparisons, percentages of horses in each classification were determined.

### 2.5. Statistical Analysis

All analyses were performed using SAS 9.4 (Cary, NC) and a significance threshold of *p* < 0.05 was used. Descriptive statistics for cranial to caudal and dorsal to ventral dimensions of the stomach total at all time points and with fluid post-water were determined. Percentages for qualitative comparisons between timepoints were calculated. Most cranial and most caudal ICS, height of stomach at ICS 12, and change in height of stomach at ICS 12 were analyzed with linear mixed models to account for within horse correlations of repeated timepoints per horse. Number of ICS was analyzed with a general linear mixed model as a count variable. Fixed factors of feeding status and timepoint with feeding status by timepoint interaction and a random intercept for each horse were used for linear and generalized linear mixed models. A negative binomial distribution with a natural log link was assumed for the generalized linear model. Histograms and Q-Q plots of conditional model residuals were used to evaluate the assumption of normality. For the remaining parameters, paired comparisons were performed. Thus, comparisons between unfed and fed included only the five horses that were used for both trials whereas comparisons between timepoints within a trial (unfed or fed) included all horses where the comparison could be made (up to 10 horses for unfed and up to five horses for fed comparisons). Paired *t*-tests were used to test for differences between unfed and fed for each timepoint and between timepoints for fed and unfed animals separately in change in height of stomach at ICS 12 from previous timepoint(s). Exact McNemar’s tests were used to test for differences between unfed and fed for each timepoint and between timepoints for fed and unfed animals separately for binary variables (able to image stomach; more cranial, more caudal, in more ICS compared to previous timepoint(s)). Signed rank tests were used to test for differences between unfed and fed for each timepoint and between timepoints for fed and unfed animals separately for ordinal variables (more, unchanged, or less cranial or caudal; in more, unchanged, or less ICS; increased, no change, or decreased height compared to previous timepoint(s)).

## 3. Results

All horses tolerated the study procedures well. All data are available in Appendix A. Reflux was only obtained in one horse (0.5 L) in the first trial and none in the second trial. In unfed horses, the stomach was able to be imaged in 2/10 pre-NGT, 8/10 post-NGT, 7/10 post-reflux, and 10/10 post-water. It was not able to be imaged as frequently in unfed horses pre-NGT compared to post-NGT (*p* = 0.0313) and post-water (*p* = 0.0078). In fed horses, the stomach could be imaged in 4/5 pre-NGT, 5/5 post-NGT, and 5/5 post-reflux. There were no differences between timepoints in fed horses.

The cranial to caudal and dorsal to ventral ultrasonographic dimensions of the stomach are shown in Table 1. The most cranial ICS was lower in the unfed horses post-water, than post-NGT (*p* = 0.0059), and post-reflux (*p* = 0.0017). No differences in the most cranial ICS were identified between timepoints in fed horse or between fed and unfed horses at any timepoint. The most caudal ICS was higher in the unfed horses post-water, than pre-NGT (*p* = 0.0013), and post-NGT (*p* = 0.0395,) and post-reflux than in pre-NGT (*p* = 0.0141). No differences in the most caudal ICS were identified between timepoints in the fed horses. The most caudal ICS was higher in fed than unfed horses pre-NGT (*p* = 0.0003) and post-NGT (*p* = 0.0041). There were no differences in the number of ICS the stomach could be imaged in between timepoints in fed or unfed horses. The number of ICS the stomach could be imaged in was higher in fed than unfed horses pre-NGT (*p* < 0.0001), post-NGT (*p* = 0.001), and post-reflux (*p* = 0.0018). There were no differences in the height of the stomach at ICS 12 between timepoints in fed or unfed horses. The height of the stomach at ICS 12 was higher in the fed than unfed horses post-reflux (*p* = 0.0433).

Fluid could be imaged in the stomach with ultrasound in all horses after 6 L of water was administered via NGT. The mean of the most cranial ICS fluid could be imaged in was 7.2 ± 1.4 (range 6–10), the mean of the most caudal ICS fluid could be imaged in was 11.6 ± 1.3 (range 9–13), and the mean number of ICS that fluid could be imaged in was 5.3 ± 1.7 (range 2–7).

Qualitative changes in stomach dimensions at later timepoints relative to pre-NGT, post-NGT, and post-reflux, within each trial (unfed and fed) are shown in Table 2, Table 3 and Table 4, respectively. No differences in proportions of qualitative changes in stomach dimension from pre-NGT were identified in the fed or unfed horses. In the unfed horses, the stomach could be imaged in a more cranial ICS than post-NGT more frequently post-water than post-reflux (more/no change/less *p* = 0.0078; yes/no *p* = 0.0156). No differences in proportions of changes to most caudal ICS, the number of ICS, or height at ICS 12 from post-NGT in the unfed horses were identified. No differences in proportions of qualitative changes in stomach dimension from pre-NGT were identified in the fed or unfed horses.

## 4. Discussion

The findings of this study overall support our hypothesis that feeding, nasogastric intubation, checking for nasogastric reflux, and the administration of fluid via the nasogastric tube increase the cranial to caudal and dorsal to ventral ultrasound dimensions of the stomach. The fed horses’ stomachs appeared over a larger area in both the cranial to caudal dimension, going further caudal and spanning a larger number of ICS, and the dorsal to ventral dimension, with increased height at ICS 12, than the unfed horses at variable timepoints. Looking at qualitative changes, the placement of an NGT resulted in the stomach being identified more cranially, more caudally, and over more ICS in the majority of fed and unfed horses and increased height at ICS 12. These qualitative changes are consistent with the changes seen in mean ICS parameters and height at ICS 12 measurements, but were not statistically significant. Looking at qualitative changes, checking for reflux did not result in consistent changes in ultrasound dimensions of the stomach compared to after placement of an NGT. However, compared to the pre-placement of NGT, the majority remained more cranial and in more ICS in the unfed and fed horses, more caudal in the unfed horses, and had increased height at ICS in the fed horses. These qualitative changes are consistent with the changes seen in mean ICS parameters and height at ICS 12 measurements, but were only statistically significant when comparing the most caudal ICS in the unfed horses between pre-NGT and post-reflux. The administration of 6 L of water resulting in the stomach being imaged more cranially and in more ICS in 90–100% of the horses compared to all previous timepoints and more caudally and an increased height in at least 60% of horses compared to all previous timepoints. These qualitative changes are consistent with the changes seen in mean ICS parameters and height at ICS 12 measurements, but were only statistically significant in the unfed horse comparisons to post-water for the most cranial ICS at post-NGT and post-reflux timepoints and the most caudal ICS pre-NGT and post-NGT timepoints.

The cranial to caudal location and the length of the stomach referenced to ICS is similar to previous reports [10,17,18]. In adult horses, reported cranial to caudal dimensions of the stomach are variable with the most cranial ICS at the 6–10th ICS, the most caudal ICS at 10–15th ICS, and being imaged over 3–5 ICS [10,17,18]. At baseline, our mean most cranial and most caudal ICS in the fed horses fell within these ranges, while our mean number of ICS spanned was more than reported. Within our study, we found a large amount of variability at baseline, particularly when both the unfed and fed horses are included, with some horses in the fed group having stomachs that reached more caudal (to ICS 16) and spanned more ICS (up to nine) compared to previous reports. Ranges were not reported as consistently in previous studies, but variability in stomach location including exceeding a length of five ICS was reported to be common in the experience of an expert author of one review article [10]. Direct comparisons to previous literature related to dorsal to ventral dimensions of the stomach is limited by use of ponies in one study and minimal quantitiative data presentation in the results of both studies [15,16]. In both of the previous studies, the height was observed over multiple rib spaces and primarily shown as graphs. The height at ICS 12 found in this study do appear to be within the range of the findings in those studies. In our study, we found minimal variability in baseline height in fed horses, but we were unable to image the stomach in that location in the unfed horses and found wide ranges of heights at other timepoints in both the fed and unfed horses. In the two previous experimental studies evaluating height at ICS 12, an inability to obtain the measurement at baseline (unfed) occurred in some of the horses in one study and variability in the measurement between horses or ponies following air insufflation or the administration of water was evident [15,16]. Given this wide variability, it is hard to support a specific “normal” ICS location or height at ICS 12 for the stomach on ultrasound to be used to identify distension of the stomach. It should be noted that the number of horses used in this study is not appropriate for defining a reference range, and that addition of more horses could result in a more narrow range. However, outliers would likely still represent a problem for clinical use.

Inability to visualize the stomach in the typical location in fasted horses has been previously reported [21]. It is important to note that with only two horses where the stomach could be imaged at baseline in the unfed group, this severely limited the number of data points for statistical comparison of most cranial and most caudal ICS at this timepoint. Thus, comparisons between baseline and other timepoints, and between fed and unfed at baseline for these parameters, should be interpreted very cautiously.

The differences between the fed and unfed horses in stomach ultrasound findings identified in this study are logical and consistent with previous reports [15,21]. The increases in the dimensions of the stomach on ultrasound in the fed horses compared to the unfed horses suggests that these parameters do have some association with stomach fill. Additionally, this finding is important to consider when using ultrasound to assess the stomach in horses with acute colic that have only recently stopped eating. It would have been ideal to standardize the protocols for feed withholding and feeding for the two trials. During feed withholding, horses could have been muzzled to assure no feed or shavings were consumed. During feeding, recording the amount of feed horses had consumed in relation to the time the ultrasound was performed would have been useful. However, our findings suggest that the protocols achieved the goal of simulating natural unfed and fed conditions.

The changes in gastric ultrasound dimensions identified in this study following NGT placement were suggested by findings of a previous study evaluating the effect of an indwelling NGT for 30–80 min [21]. This study provides more detail of the changes in dimension that were observed. The increase in dimensions of the stomach following NGT placement are consistent with our hypothesis based on clinical observations. It seems most likely that this is the result of air entering the stomach, which has been previously shown to alter gastric ultrasound [15]. This finding is important to consider when using ultrasound to assess the stomach in horses with colic that have or have had a NGT in place. In the previous study, following air insufflation, the height of the stomach at ICS 12 decreased over the 4 h post insufflation [15].

In this study, checking for reflux did not consistently decrease the gastric ultrasound dimensions. This finding suggests that gastric ultrasound dimensions may not be a reliable way to determine if gastric reflux has been effectively removed. It should be noted that we only created a siphon on one occasion and applied suction relatively briefly in this study when checking for reflux. In clinical cases, more attempts may be made to retrieve reflux and more suction may be applied to remove gas. This is supported by findings in a clinical study of horses with gastric reflux where removal of reflux only decreased the number of ICS the stomach could be imaged in by −27%, and the stomach would still have been classified as enlarged if spanning ≥ 6 ICS in 14/52 examinations [6].

Fluid administration also increased the dimensions of the stomach compared to previous timepoints as expected based on a previous study of NGT fluid administration [16]. This would make it seem that an increase in ultrasonographic dimensions of the stomach could be a reliable method to detect fluid distension. However, not all changes were statistically different from previous timepoints and not all horses had increases across all parameters. The overlap in the reported ranges of the parameters between makes defining a cutoff value challenging. Further, as noted, statistical comparisons for some of the parameters to baseline are limited by only being able to visualize the stomach in two horses in the unfed group at that timepoint. Given the qualitative changes that occurred, there may be potential for serial measurements to be more useful, particularly if horses are not being fed and the NGT is capped and not being manipulated. Another potentially useful parameter would be the visualization of fluid within the stomach. In our study, we were able to consistently identify fluid in the stomach following administration of 6 L of water. This is consistent with the study in clinical cases of horses with reflux where they found the finding that correlated best with amount of reflux obtained was the number of ICS where fluid could be visualized with ultrasound [6]. These findings indicate that further evaluation of clinical cases focused on the identification of fluid in the stomach with ultrasound is warranted.

This study has several limitations. Only a small number of horses were evaluated, particularly in the fed trial. Additionally, the horses in this study were clinically healthy. Caution should be used when applying the findings of this study to horses with colic. Not all possible ultrasonographic parameters were evaluated in this study. We selected those used most frequently clinically (cranial to caudal dimensions references to ICS) and elected to evaluate height at ICS 12 based on two experimental studies, supporting increases associated with air insufflation and water administration [15,16]

## 5. Conclusions

The findings of this study overall support our hypothesis that feeding, nasogastric intubation, checking for nasogastric reflux, and the administration of fluid via the nasogastric tube increase the cranial to caudal and dorsal to ventral ultrasound dimensions of the stomach. As a result, using the ultrasonographic location and the size of the stomach in horses that have had been recently fed and/or had a NGT placed to identify gastric distension, can result in the misdiagnosis of distension. However, because the administration of 6 L of water resulted in the consistent identification of fluid in the stomach on ultrasound, this finding may be a more useful parameter clinically to identify horses likely to have significant gastric reflux.

## Figures and Tables

**Table 1 animals-12-03433-t001:** Cranial to caudal and dorsal to ventral ultrasonographic dimensions of the stomach.

	Most Cranial ICS	Most Caudal ICS	Number of ICS (Inclusive)	Height at ICS 12 (cm)
Unfed Pre-NGT	8 ± 2.8 ^a,b^ (6–10) *n* = 2	8.5 ± 2.1 *^,a^ (7–10) *n* = 2	0.3 ± 0.7 * (0–2) *n* = 10	Unable to image stomach in ICS 12 in any horse
Unfed Post-NGT	8.6 ± 1.1 ^a^ (7–10) *n* = 8	12.3 ± 2.4 *^,a,b^ (8–15) *n* = 8	3.7 ± 2.8 * (0–8) *n* = 10	26.7 ± 7.9 (18.5–38.5) *n* = 5
Unfed Post-reflux	8.9 ± 2.0 ^a^ (6–11) *n* = 7	13.3 ± 2.1 ^b,c^ (10–16) *n* = 7	3.8 ± 3.2 * (0–9) *n* = 10	25.8 ± 4.1 * (20.0–32.5) *n* = 6
Unfed Post-water	6.4 ± 1.3 ^b^ (4–9) *n* = 10	14.4 ± 1.6 ^c^ (12–16) *n* = 10	9.0 ± 1.8 (7–12) *n* = 10	28.5 ± 3.5 (22.5–32.5) *n* = 10
Fed Pre-NGT	7.5 ± 1.0 (6–8) *n* = 4	14.5 ± 1.0 * (14–16) *n* = 4	6.4 ± 3.7 * (0–9) *n* = 5	28.5 ± 0.6 (28.0–29.0) *n* = 4
Fed Post-NGT	7.6 ± 1.9 (6–11) *n* = 5	14.8 ± 0.8 * (14–16) *n* = 5	8.2 ± 2.5 * (4–10) *n* = 5	29.4 ± 1.7 (27.0–31.0) *n* = 5
Fed Post-reflux	7.8 ± 2.5 (6–12) *n* = 5	14.8 ± 0.8 (14–16) *n* = 5	8.0 ± 2.3 * (4–10) *n* = 5	28.7 ± 3.9 * (24.0–34.0) *n* = 5

ICS = Intercostal space. For most cranial and most caudal ICS, the values given are the most cranial or most caudal ICS that the stomach could be visualized with ultrasound. For ICS spanned, the value given is the number of ICS over which the stomach could be imaged inclusive of the most cranial and most caudal ICS. Height at ICS 12 = distance from the level of the point of the elbow to the most dorsal point the stomach could be imaged in ICS 12. NGT = nasogastic tube. Pre-NGT data were taken at baseline, before the NGT was placed. Post-NGT data were taken within 5 min of the NGT begin placed. Post-reflux data were taken within 5 min of checking for reflux. Post-water data were taken within 5 min of administering 6 L of water via the NGT. Data presented as mean ± SD (range). *n* = number of horses able to record finding. Significant differences between fed and unfed at a timepoint are denoted by *. Significant difference between timepoints in unfed horses are denoted by letter superscripts. No significant differences between timepoints were identified in fed horses.

**Table 2 animals-12-03433-t002:** Qualitative changes in stomach dimensions from pre-NGT dimensions (expressed as percentages of horses with each change).

	Most Cranial ICS		Most Caudal ICS		Number ICS (Inclusive)		Height at ICS 12
	M:NC:L	More Cranial Y:N	M:NC:L	More Caudal Y:N	M:NC:L	More ICS Y:N	I:NC:D
Unfed Post-NGT	70:20:10	70:30	80:20:0	80:20	80:20:0	80:20	Unable to determine: stomach not visible in ICS 12 pre-NGT
Unfed Post-reflux	60:30:10	60:40	70:30:0	70:30	60:40:0	60:40	
Unfed Post-water	90:10:0	90:10	100:0:0	100:0	100:0:0	100:0	
Fed Post-NGT	80:20:0	80:20	60:40:0	60:40	100:0:0	100:0	75:25:0
Fed Post-Reflux	60:40:0	60:40	40:60:0	40:60	80:20:0	80:20	75:0:25

ICS = Intercostal space. For most cranial and most caudal ICS, the values given are the most cranial or most caudal ICS that the stomach could be visualized with ultrasound. For ICS spanned, the value given is the number of ICS over which the stomach could be imaged inclusive of the most cranial and most caudal ICS. Height at ICS 12 = distance from the level of the point of the elbow to the most dorsal point the stomach could be imaged in ICS 12. NGT = nasogastic tube. Pre-NGT data were taken at baseline, before the NGT was placed. Post-NGT data were taken within 5 min of the NGT begin placed. Post-reflux data were taken within 5 min of checking for reflux. Post-water data were taken within 5 min of administering 6 L of water via the NGT. Unfed timepoints changes relative to unfed pre-NGT and fed timepoints changes relative to fed pre-NGT. Data presented as percentages. *n* = number of horses recorded. *n* = 10 for all unfed comparisons; *n* = 5 for fed ICS comparisons; *n* = 4 for fed height at ICS 12 comparisons. M = more, NC = no change, L = less, I = increased, D = decreased, Y = yes, N = no.

**Table 3 animals-12-03433-t003:** Qualitative changes in stomach dimensions from post-NGT dimensions (expressed as percentages of horses with each change).

	Most Cranial ICS		Most Caudal ICS		Number ICS (Inclusive)		Height at ICS 12
	M:NC:L	More Cranial Y:N	M:NC:L	More Caudal Y:N	M:NC:L	More ICS Y:N	I:NC:D
Unfed Post-reflux	20:40:40 *	20:80 *	50:30:20	50:50	50:20:30	50:50	40:0:60
Unfed Post-water	90:10:0 *	90:10 *	80:10:10	80:20	100:0:0	100:0	60:0:40
Fed Post-reflux	20:40:40	20:80	20:60:20	20:80	20:40:40	20:80	40:0:60

ICS = Intercostal space. For most cranial and most caudal ICS, the values given are the most cranial or most caudal ICS that the stomach could be visualized with ultrasound. For ICS spanned, the value given is the number of ICS over which the stomach could be imaged inclusive of the most cranial and most caudal ICS. Height at ICS 12 = distance from the level of the point of the elbow to the most dorsal point the stomach could be imaged in ICS 12. NGT = nasogastic tube. Post-NGT data were taken within 5 min of the NGT begin placed. Post-reflux data were taken within 5 min of checking for reflux. Post-water data were taken within 5 min of administering 6 L of water via the NGT. Unfed timepoint changes relative to unfed post-NGT and fed timepoints changes relative to fed post-NGT. Data presented as percentages. *n* = number of horses evaluated. *n* = 10 for unfed ICS comparisons; *n* = 5 for fed ICS comparisons; *n* = 5 for unfed and fed height at ICS 12 comparisons. M = more, NC = no change, L = less, I = increased, D = decreased, Y = yes, N = no; Significant difference in proportions between post-reflux and post-water unfed horses denoted by *.

**Table 4 animals-12-03433-t004:** Qualitative changes in stomach dimensions from post-reflux dimensions (expressed as percentages of horses with each change).

	Most Cranial ICS		Most Caudal ICS		Number ICS (Inclusive)		Height at ICS 12
	M:NC:L	More Cranial Y:N	M:NC:L	More Caudal Y:N	M:NC:L	More ICS Y:N	I:NC:D
Unfed Post-water	100:0:0	100:0	60:30:10	60:40	90:10:0	90:10	60:40:0

ICS = Intercostal space. For most cranial and most caudal ICS, the values given are the most cranial or most caudal ICS that the stomach could be visualized with ultrasound. For ICS spanned, the value given is the number of ICS over which the stomach could be imaged inclusive of the most cranial and most caudal ICS. Height at ICS 12 = distance from the level of the point of the elbow to the most dorsal point the stomach could be imaged in ICS 12. NGT = nasogastic tube. Post-reflux data were taken within 5 min of checking for reflux. Post-water data were taken within 5 min of administering 6 L of water via the NGT. Changes relative to unfed post-reflux. Data presented as percentages. *n* = number of horses evaluated. *n*= 10 for ICS comparison; N = 5 for height at ICS 12 comparisons. M = more, NC = no change, L = less, I = increased, D = decreased, Y = yes, N = no.

## Data Availability

Data is contained within Appendix A.

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
