# Peer review of "Effect of Nasogastric Tube Placement, Manipulation, and Fluid Administration on Transcutaneous Ultrasound Visualization and Assessment of Stomach Position in Healthy Unfed and Fed Horses"

_animals, 2022, doi:10.3390/ani12233433_

Round 1

Reviewer 1 Report

General comments:

Overall, this is an interesting study with potential clinical application. Throughout the manuscript, there were some issues with sentence structure that made it a bit difficult to read. More detail could be provided in the results related to the ultrasound findings. Could an overall length and/or height of the stomach (ICSa to ICSb) be presented? Is it similar to what is already described in the literature? The data measurements at the different timepoints are not presented at all?

Line by line review:

Abstract

Lines 19-21: comma needed between “passed” and “the”

Line 26-27: remove “and” after tube placement

Line 33-34: nasogastric tube placed, add placed

Introduction

Line 40-42: Vague sentence, please be more explicit. Not exactly sure what the authors are suggesting

Line 69-70: Please make the sentence starting with, “Despite the frequency of inclusion…”more concise

Line 101-103: The hypothesis is a bit vague. Can it be written more directly? What “changes” were expected? Increase in dimension cranial to caudal and dorsal to ventral?

Materials and Methods

2.2 Study Design: Was the amount of fee that the horse consumed measured? Approximately how much consumed in grain and hay? What does “fed normally” mean? Free choice hay? Is it possible that a horse did not eat? Or ate shavings? How much of the NGT (approximately) was left in the stomach? Could that impact the findings? How soon after feeding was the NGT placed?

2.4 Data collected: Was there a reason why the dorsal to ventral measurements were not measured after fluid administration?

Line 189: tube?

Results

Line 212: p value missing period (p=00003)

Table 1: It is appreciated that the stomach could only be identified in 2 horses pre-NGT, but this limits what statistics can be performed with the data obtained from these animals.

Line 225: Sentence structure needs to be addressed. “The most cranial ICS fluid could be imaged in was 7.2…” in what? Same sentence structure issues in  remaining sentences of this paragraph. What are the units? Same issue in Table 1, what units are being used?

Line 229: remove ‘in”…within in

Table 2: It is unclear what the numbers represent at first glance. I believe a clearer description of what data is being present is needed in methods. Furthermore, Table title may be changed to Percent change in stomach dimensions… to make it more obvious to the reader.

Table 3 and 4: Same comment as for Table 2

Discussion:

The first paragraph repeats what was presented in the results.

The statement, “The cranial to caudal location of the stomach references to ICS and height at ICS 12 found in this study were similar to previous reports,” is not supported by the data provided. No data indicating location is actually presented. A better discussion of expected and unexpected findings and why are needed.

Conclusion: Clinical significance of last sentence in this section is needed. Please be more explicit. It is unclear what the clinical application of this study’s findings are.

Reviewer 2 Report

Dear authors,

I have no major remarks on your manuscript.

The only suggestion I could make, although not essential, would be to do a figure with a horse and place marks around the rib/thorax area where one should expect to find the stomach on ultrasound, with changes in sizes of course.

Author Response

The authors thank the reviewer for their time in reviewing the manuscript.  At this time, we are unable to produce the type of figure suggested within the response to reviewer timeline of the journal and without the aid of a medical illustrator.  We would be happy to provide additional figures/graphs if needed given more time.

Reviewer 3 Report

Reviewer comments for manuscript ID animals-2029942 entitled ‘Effect of nasogastric tube placement, manipulation, and fluid 2 administration on transcutaneous ultrasound visualization and assessment of stomach position in healthy unfed and fed horses’

General Comments

It is relevant study on the gastric ultrasonography in horses that is frequently being used in equine patients especially in colic cases. Specific and faster techniques like ‘FLASH - Fast Localised Abdominal Sonography of Horses’ have improved diagnosis and hence better treatment outcomes in abdominal crisis. Assessment of the size of the cranio-caudal and dorso-ventral aspects of the stomach during ultrasonography demonstrated in this study should aid in the diagnosis and prognosis in patients with colic. However, I could not identify any unique finding in the study that is of significance to the present literature on the subject. Results and statistical methods are well written whereas the introduction is mundane. Discussion is very vague and superficial and mostly repetition of the results and limited to comparison with previous works. I suggest a more effort on these weak aspects of the manuscript.

Specific Comments

Lines 14-17: I am sorry I am not able to understand these lines. Please clarify.

Lines 16-18: Please reframe as ‘In a group of healthy horses, ultrasonography revealed that feeding, stomach tube placement, and giving fluid, increased the size of the stomach.

Line 18-19: This sentence is ambiguous. Please reword it.

Line 34-35: Does this mean that identification of fluid in stomach by ultrasound is more important diagnostic parameter than the increase in size of the stomach? Please clarify and rewrite.

Line 43: Please refer to the “FLASH” technique of diagnostic ultrasound in colic horses.

Lines 259-61: What could be the possible reason for this ? Please clarify.

Line 259-77: These are simply reporting of results and hence a repetition. Discussion is missing.

Round 2

Reviewer 3 Report

Reviewer comments for manuscript Id animals-2029942 entitled ‘Effect of nasogastric tube placement, manipulation, and fluid administration on transcutaneous ultrasound visualization and assessment of stomach position in healthy unfed and fed horses’

General comments

I congratulate the authors for their effort to improve the manuscript through the corrections suggested. The manuscript is much improved. It should be able to add some knowledge to the ever-expanding work on equine ultrasonography. I am happy with the corrections done though I still have reservation on the novelty of the research.

Specific comments

Line 16: Please replace ‘effects’ with ‘affects’